# Spatiotemporal Distribution of Drought and Humidity in China Based on the Pedj Drought Index (PDI)

**Xu Wu [1], Xiaojing Shen [1,2,3] and Jianshe Li [1,\*]**

[1] School of Civil and Hydraulic Engineering, Ningxia University, Yinchuan 750021, China; wxmqhz@163.com (X.W.); sxj15191418250@163.com (X.S.)

[2] Engineering Technology Research Center of Water-Saving and Water Resource Regulation in Ningxia, Ningxia University, Yinchuan 750021, China

[3] Engineering Research Center for Efficient Utilization of Modern Agricultural Water Resources in Arid Regions, Ministry of Education, Ningxia University, Yinchuan 750021, China

\* Correspondence: ljs292849539@163.com

**Abstract:** Drought is one of the most devastating natural disasters, especially in China. In drought assessment, the PDI has high robustness, is easier to obtain than indices such as SPEI and PDSI, and is more advantageous in regions with sparse stations. The present study employs the annual PDI with the precipitation and temperature data from 830 meteorological stations to systematically study the interannual variation characteristics of drought and humidity in China during 1970–2019. The results showed the following: (1) 26.6% of the total statistics from meteorological stations showed significant ($p < 0.05$) increases in annual PDI values throughout China during 1970–2019. (2) Air temperature plays a more remarkable role than precipitation in assessing the drying trend with PDI throughout China. (3) About 71% of stations experienced more drought events (PDI > 1) than humidity events (PDI < −1), 14% of stations experienced more humidity events than drought events. (5) All stations experienced drought events (PDI > 1) with a frequency range from 6% to 32% and humidity events (PDI < −1) with a frequency range from 8% to 36%. Most of the stations experienced extreme drought events (PDI > 3) with a frequency range from 2% to 10%, while only 177 stations experienced extreme humidity events (PDI < −3) with a frequency range from 2% to 4%. (6) More than 67% of stations experienced drought conditions during both periods of 1972–1974 and 2000–2002 and even exceeded 80% in the three years 1972, 2000, and 2001. Both periods of 1976–1983 and 1985–1999 can be considered to be a humidity period throughout China. In conclusion, the PDI successfully expresses the interannual variation characteristics of drought and humidity events throughout China previously captured by other prominent, recommended drought indices.

**Keywords:** Pedj Drought Index (PDI); drought; humidity; China

## 1. Introduction

Drought is one of the most devastating natural disasters [1–3] and is directly associated with geographical location, altitude, and the distance from each big river system. The term drought is differently characterized according to research characteristics [4]. Drought is generally defined a significant shortage of natural freshwater supplies over a long period of time due to changes in precipitation and temperature patterns [5]. In recent years, the increasing human activities and climate warming have resulted in a change in the spatial distribution pattern of drought [6,7]. Several drought indices have already been adopted to evaluate various drought characteristics [8]. The more common drought indices are Palmer Drought Severity Index (PDSI) (Palmer, 1965), Standardized Precipitation Evapotranspiration Index (SPEI) [9], Standardized Precipitation Index (SPI) [10], Reclamation Drought Index (RDI) [11], Moisture Anomaly Index (Z-index) [12], and Precipitation Anomaly Index (RAI) [13]. Precipitation is the only indispensable input hydrometeorological variable in these indices.

China is located in eastern Eurasia, the Pacific west bank, and has a monsoon climate. The precipitation has obvious seasonal change, which leads to differences in the spatiotemporal distribution of drought. For the past few years, the rapid spread of drought has occurred in southwest China. The extreme drought events that happened in southwest China during 2009–2010 were significantly associated with a significantly high temperature and scarce rainfall [14–17]. Zou et al. [18] calculated the Palmer Drought Severity Index (PDSI) with the monthly temperature and rainfall data during 1951–2003 and discovered that the drought areas in North China had significantly increased since the late 1990s. The research of Lu et al. [19–21] (2008; 2013; 2016) demonstrated that, in the past few decades, the Liaohe Plain, the Haihe Plain, the Loess Plateau, the Szechwan Basin, and the Yunnan–Guizhou Plateau formed a serious drying strip, with increasing drought frequency, which is closely related to the decrease of precipitation. Moreover, severe and extreme droughts have become more serious since the late 1990s for all of China, with the dry area increasing by 3.72% per decade, especially in North China, Northeast China, and western Northwest China [3]. Meanwhile, Xu et al. [22] analyzed the drought characteristic in China during 1960–2012 with a three-dimensional clustering method. The result shown that, in the past half century, the most severe drought occurred during 1962–1963 and 2010–2011 with a widely drought affected area. Ma et al. [23] analyzed all meteorological drought events in China from 1961 to 2017 using the Standard Precipitation Evapotranspiration Index (SPEI), which revealed that the frequency of drought events in the eastern monsoon area of China is high, the duration is short, and the intensity is weak, while this was reversed in the northwest arid region.

In previous studies, different drought indices, such as the Palmer Drought Severity Index (PDSI), Standard Precipitation Index (SPI), Standard Precipitation Evapotranspiration Index (SPEI), Percent of Normal (PN), Standardized Precipitation Index (SPI), China-Z Index (CZI), and Declines Index (DI), were widely used to evaluate the drought characteristic in China [3,21,23,24]. Each has its own characteristics and has achieved fruitful results in drought/humidity research in China. The ability of a drought index to assess drought severity involves robustness, tractability, transparency, sophistication, extendibility, and dimensionality, among which robustness and treatability are the most basic and important abilities [1]. The robustness and treatability include not only the spatiotemporal comparability of the drought index but also the manageability of the practical aspects of drought index. Furthermore, drought indices should be calculated using readily available data as much as possible to reflect drought conditions most intuitively [25]. Hence, SPI was recommended for evaluating drought characteristics due to its acceptable robustness and treatability for detecting drought [1,22]. Although readily available and popular, the sensitivity of the SPI in assessing drought is closely related to the soil moisture and groundwater. Meanwhile, the assessment ignores temperature, an important meteorological factor. These factors greatly reduce the SPI's ability to assess droughts in regions with extensive arid and semi-arid areas [4,26,27].

In contrast, the Pedj Drought Index (PDI) is sensitive to both precipitation and temperature, with high performance for assessing the onset of drought over extended periods (annual) [25]. In previous research, the PDI has been reported to identify drought events in a similar way to the well-known United Nations Environment Programme [28] Drought Index (AI) [25,29]. Another bright feature of the PDI is its outstanding ability to evaluate drought characteristics in regions with sparse meteorological stations and poorly recorded data [29]. In summary, the PDI has high robustness in drought assessment and is easier to obtain than SPEI, PDSI, etc., and has more advantages in drought assessment in regions with sparse stations. Therefore, the PDI is efficient and reliable in assessing drought characteristics. However, the adaptability of the PDI in spatiotemporal drought variability research throughout China is yet to be reported. The present article aims to fill this research gap.

The present article employs the annual PDI with the precipitation and temperature data from 830 meteorological stations to systematically study the interannual variation

characteristics of drought and humidity in China during 1970–2019. The specific objectives are as follows: (1) identify the drought and humidity severities of stations with statistically significant ($p < 0.05$) trends in the past 50 years in China; (2) analyze the impacts of temperature and precipitation changes on such significant trends; (3) calculate the frequency, longest duration, and spatial extent of both drought and humidity events throughout China.

## 2. Study Area and Data

### 2.1. Study Area

The People's Republic of China is located in the east of the Asian continent and on the west coast of the Pacific Ocean, with a land area of 9.6 million km$^2$, extending from latitude of about 3°51′ N to 53°33′ N and from longitude of about 73°33′ E to 135°05′ E. The proportion of the landforms in China is as follows: mountain area 33.3%, plateau area 26%, basin area 18.8%, plain area 12%, and hilly area 9.9%, which makes it complex and diverse.

China has a vast territory, spanning five climatic zones from south to north: tropical zone, subtropical zone, warm temperate zone, middle temperate zone, and cold temperate zone [18]. There are multiple climate types in China, e.g., temperate monsoon climate, subtropical monsoon climate, tropical monsoon climate, tropical rain forest climate, temperate continental climate, and plateau mountain climate. Since the East Asian monsoon climate zone covers the vast majority of continental China, which is wet in summer and dry in winter, the annual distribution of precipitation is obviously different and the interannual variation is very significant [3,30]. Meanwhile, there is a significant negative correlation between precipitation and temperature [22].

Due to differences in precipitation (data of 830 meteorological stations during 1970–2019), China is divided into three main regions: humid region, transition region (sub-humid and semi-arid region), and arid region (Figure 1), accounting for 47%, 29%, and 24% of the total area, respectively. Because it is deeply influenced by the East Asian monsoon, southeastern China (humid region) is dominated by a warm and humid climate and has a relatively wetter climate, with annual precipitation ranging from 800 to 2200 mm and temperature ranging from 12.5 to 22 °C [18]. In the transition region, the precipitation (400 to 800 mm) is reduced compared with that in the humid region due to the weakening of the monsoon's influence, and the temperature is also reduced (5 to 16 °C). In the northwest of China, the cold and arid climate dominates much of the region, with annual precipitation ranging from 50 to 400 mm and temperature ranging from −2 to 8 °C. It should be emphasized that precipitation and temperature show a significant negative correlation with the distance from the ocean [22].

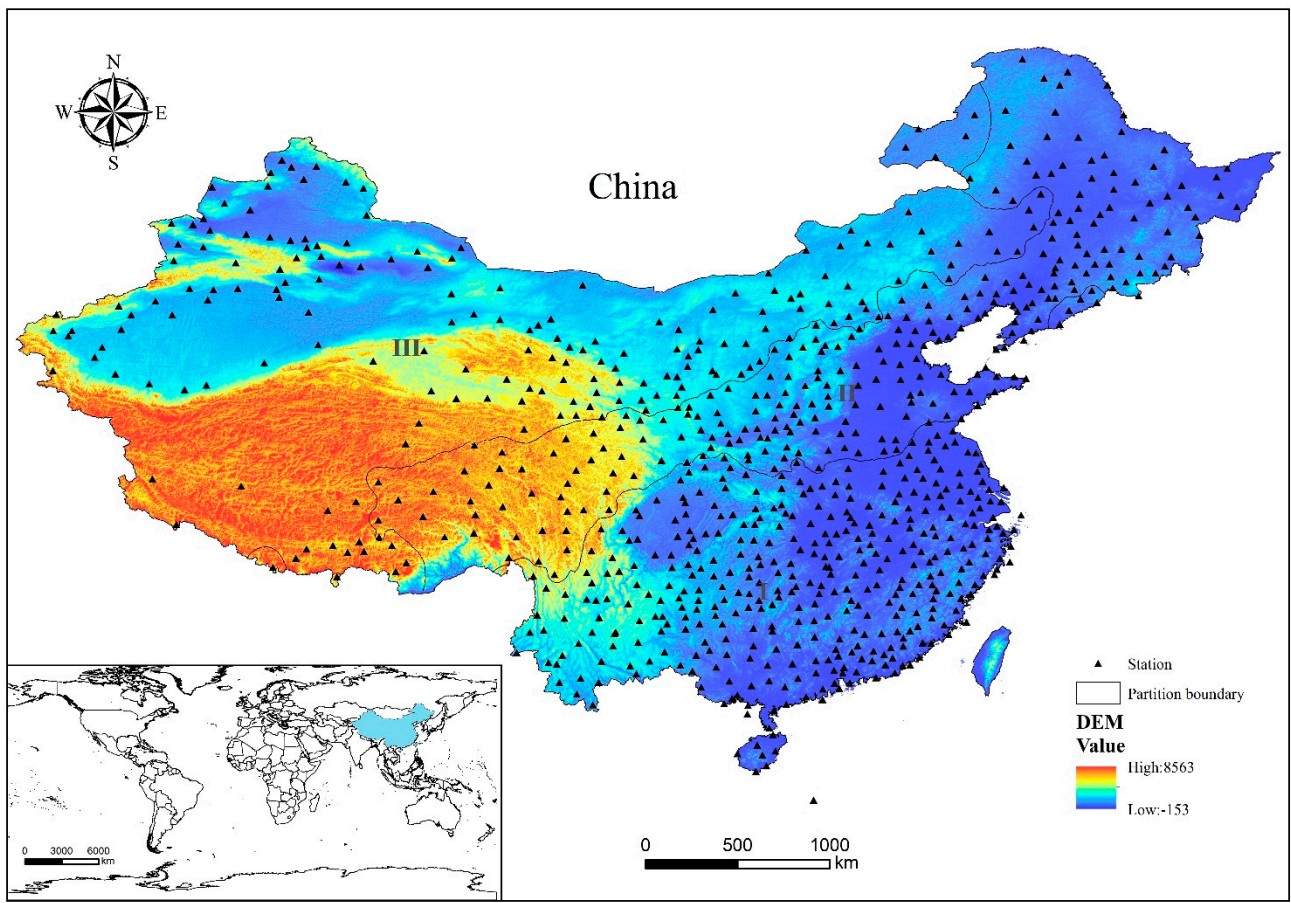

**Figure 1.** Geographical location, DEM, weather station, and sub–regions of the study area.

*2.2. Data*

Daily precipitation and air temperature data from 1970 to 2019 were acquired from the National Meteorological Science Data Center of China (http://data.cma.cn, accessed on 22 December 2021). The missing values in each station were filled with the mean values of the nearest five stations calculated with the Tyson Polygon Method.

In order to adequately analyze the research results, China was divided into three major regions: humid region (I), sub-humid and semi-arid region (II), and arid region (III), with annual precipitation of >800 mm, 400–800 mm, and <400 mm, respectively. The regional divisions were obtained based on the annual precipitation data from 830 meteorological stations with the Inverse Distance Weighting Interpolation Method, the data were interpolated into1 km gridded data (see Figure 1). There were 383, 261, and 186 meteorological stations in region I, II, and III, respectively (see Table 1).

**Table 1.** Distribution of meteorological stations selected by present research.

| Region | Climate | Number of Stations | Record Period |
|--------|---------|--------------------|----------------|
| I | Humid | 383 | 1970–2019 |
| II | Sub-humid and semi-arid | 261 | 1970–2019 |
| III | Arid | 186 | 1970–2019 |

*2.3. Calculation of PDI*

The PDI developed by Pedj [31] was selected to identify drought periods and severity with multiple spatial and temporal scales [25,29]. PDI is an indicator determined by both temperature and precipitation. Hence, two standardized anomaly indices (SAIs) need to be

calculated with priority, i.e., mean temperature and precipitation, respectively. The SAI is calculated as follows:

$$SAI = \frac{x - \bar{x}}{s} \tag{1}$$

where $x$ is a particular year record, and $\bar{x}$ and $s$ are the mean and the standard deviation of all year records over the time period, respectively. Then PDI is calculated as

$$PDI = SAI_T - SAI_P \tag{2}$$

where $SAI_T$ and $SAI_p$ are the SAIs of mean temperature and precipitation, respectively, on the corresponding time scale. In the present study, the magnitudes of drought and humidity were defined as nine types, and each standard of the PDI range is shown in Table 2.

**Table 2.** Classification standard of drought based on PDI.

| Drought/Humidity Classification | Abbreviation | PDI Range |
|---|---|---|
| Extreme drought | $D_4$ | More than 3 |
| Severe drought | $D_3$ | 2 to 3 |
| Moderate drought | $D_2$ | 1 to 2 |
| Light drought | $D_1$ | 0 to 1 |
| Normal | N | 0 |
| Light humidity | $H_1$ | −1 to 0 |
| Moderate humidity | $H_2$ | −2 to −1 |
| Severe humidity | $H_3$ | −3 to −2 |
| Extreme humidity | $H_4$ | Less than −3 |

### 2.4. Statistical Analysis Method

Sen's slope estimator [32] was used to calculate the trends of PDI, $SAI_T$, and $SAI_P$ time series. The Spearman rank correlation test [33] was used to detect the pairwise correlation between PDI, $SAI_T$, and $SAI_P$. The Spearman's rank correlation coefficient evaluates the linear relationship between two arrays, the coefficient is between −1 and +1, the negative coefficient indicates decreasing trend and vice versa. Since the test is nonparametric, the two sets of variables do not need to be normally distributed [29]. Additionally, the nonparametric Mann–Kendall (MK) [34–36] test was applied for data significance testing.

### 3. Results

Trend analysis identified 221 meteorological stations with significant ($p < 0.05$) increases in annual PDI values studied throughout China during 1970–2019, accounting 26.6% of total statistics stations. Regions I, II, and III had 121, 63, and 37 stations with significant increases in annual PDI, accounting for 31.6%, 24.1%, and 19.9%, respectively, of the number of statistical stations in their respective regions. Only 57 stations showed significant ($p < 0.05$) decreases, accounting 6.9% of total statistics stations. Regions I, II, and III had 13, 17, and 27 stations with significant decreases in annual PDI, accounting for 3.4%, 6.5%, and 14.5%, respectively, of the number of statistical stations in their respective regions (see Table 3). Among the 221 stations with increased annual PDI trends, 216 (accounting for 97.7%) showed significant increases in annual $SAI_T$, and 5 showed significant decreases in annual $SAI_T$. However, 204 (accounting for 92.3%, 204 out of 221) did not show significant trends in annual $SAI_p$, 14 showed significant increases in annual $SAI_p$, and 3 showed significant decreases in annual $SAI_P$ (see Figure 2a–c). Among the 57 stations with decreased annual PDI trends, 31 (accounting for 54.4%, 31 out of 57) showed significant decreases in annual $SAI_T$, 14 showed significant increases in annual $SAI_T$, and 12 did not show a significant trend in annual $SAI_T$. Of these 57 stations, 48 (accounting for 84.2%) showed a significant increase trend in annual $SAI_P$, while 9 did not show a significant trend in annual $SAI_P$ (see Figure 2a–c).

**Table 3.** Statistical table of stations with annual PDI trends in China selected by the presented research during 1970–2019.

| Region | Insignificant | | Significant | | | |
| --- | --- | --- | --- | --- | --- | --- |
| | Count | Percentage | Increase | | Decrease | |
| | | | Count | Percentage | Count | Percentage |
| I | 249 | 65.0 | 121 | 31.6 | 13 | 3.4 |
| II | 181 | 69.3 | 63 | 24.1 | 17 | 6.5 |
| III | 122 | 65.6 | 37 | 19.9 | 27 | 14.5 |
| Total | 552 | 66.4 | 221 | 26.6 | 57 | 6.9 |

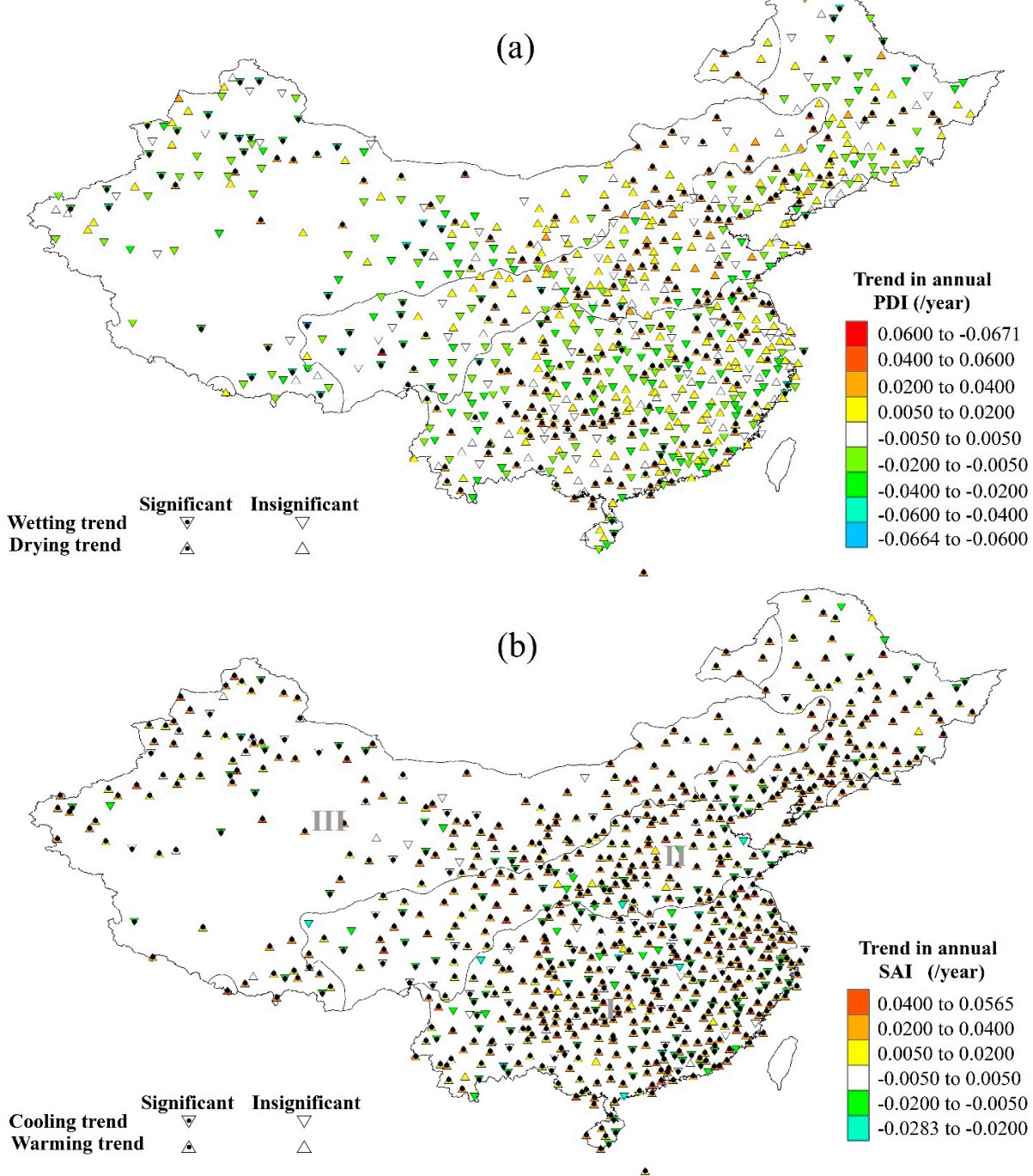

**Figure 2.** *Cont*.

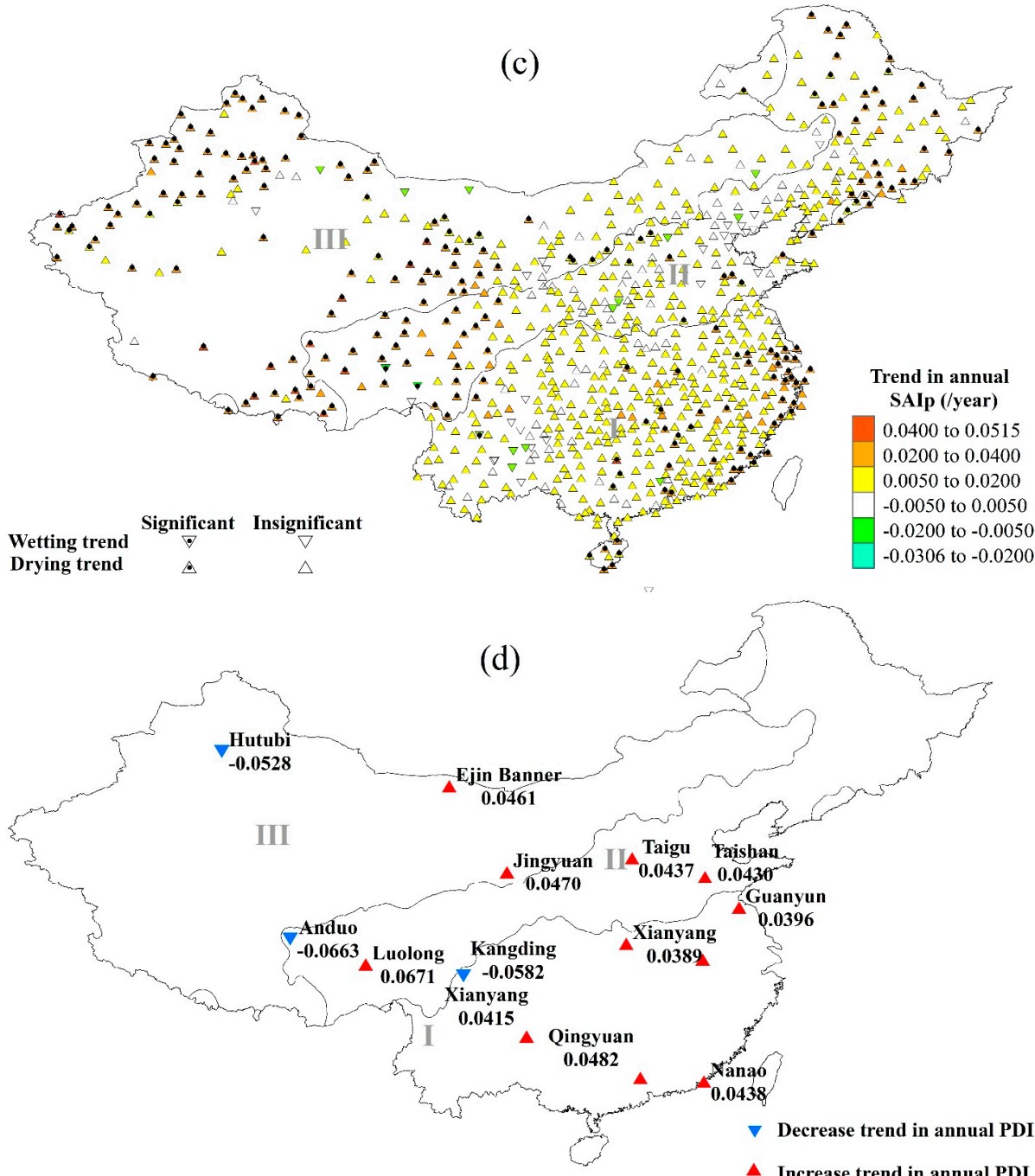

**Figure 2.** Spatial distribution maps of trends in annual (**a**) PDI, (**b**) SAI$_T$, and (**c**) SAI$_P$ at all 830 stations in China selected by the presented research during 1970–2019. (**d**) The top 5% of stations with significant changes in annual PDI in each region of China selected by the presented research during 1970–2019.

The top 5% of stations with significant changes in annual PDI in each region (see Figure 2a) were selected for extreme stations' analysis (see Figures 3 and 4). The top 5% of region I stations (6 out of 121) with significant increases in annual PDI had values of 0.0482, 0.0438, 0.0415, 0.0396, 0.0389, and 0.0386/year, respectively. The top 5% of region II stations (3 out of 63) with significant increases in annual PDI had values of 0.0671, 0.0437, and 0.0430/year, respectively. The top 5% of region III stations (2 out of 37) with significant increases in annual PDI had values of 0.0471 and 0.0462/year, respectively. Extreme stations with significant increases in annual PDI of region I had more drought

events after 2000 (see Figure 3a–f). Extreme stations of region II had more drought events since the middle 1990s (see Figure 3g–i). The one extreme station of region III had more drought events after 2000 (see Figure 3j), the other one had more drought events since the 1980s (see Figure 3k). The top 5% of region I, region II, and region III stations (1 out of 13, 1 out of 17, 1 out of 27) with significant decreases in annual PDI had values of 0.430, 0.471, and 0.462/year, respectively. Extreme stations of region I, II, and III with significant decreases in annual PDI had more humid years since the early 1990s (see Figure 4a), the middle 1980s (see Figure 4b), and the early 1980s (see Figure 4c), respectively.

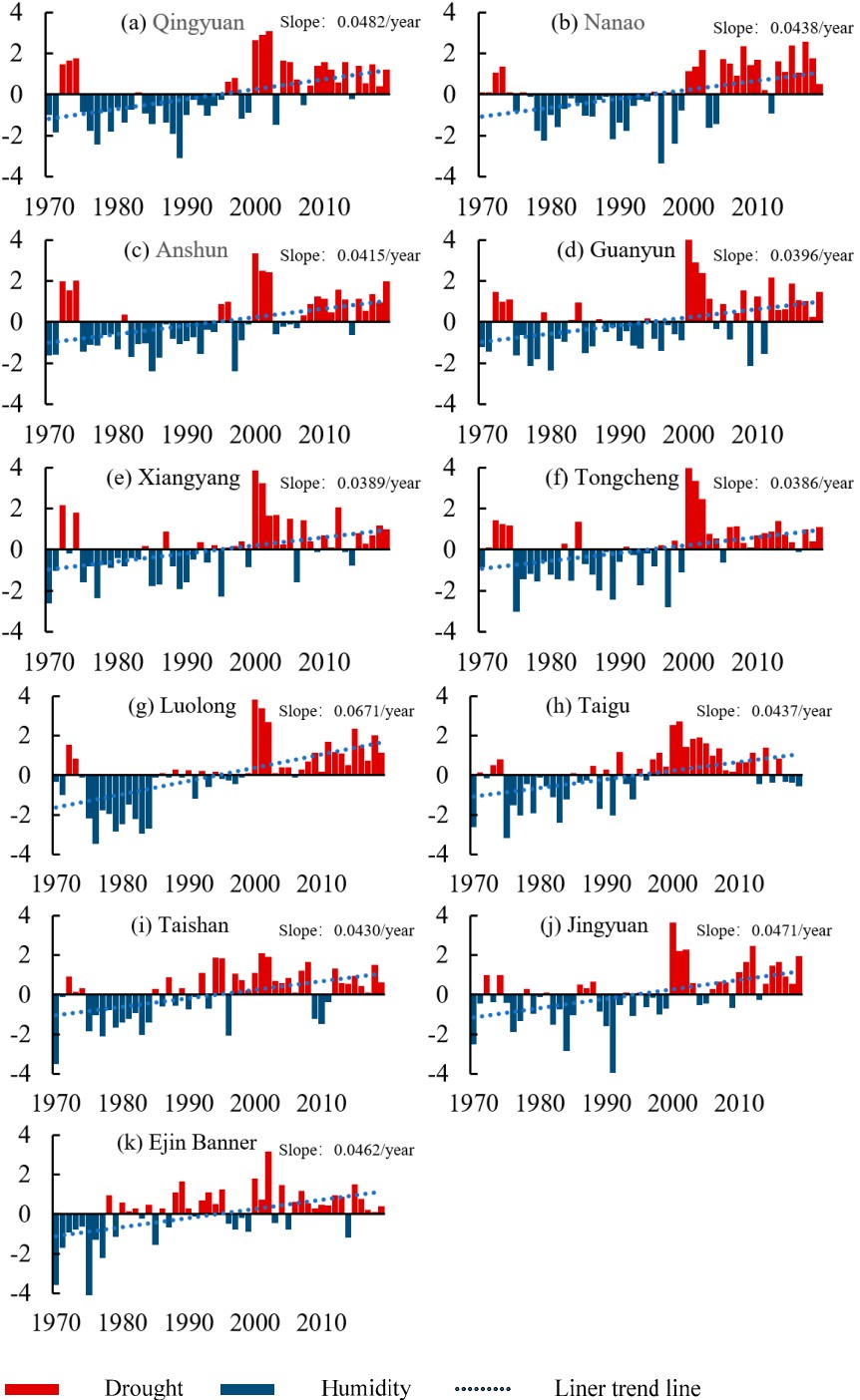

**Figure 3.** Time series of top 5% stations with significant increases in annual PDI in (**a–f**) region I, (**g–i**) region II, and (**j–k**) region III.

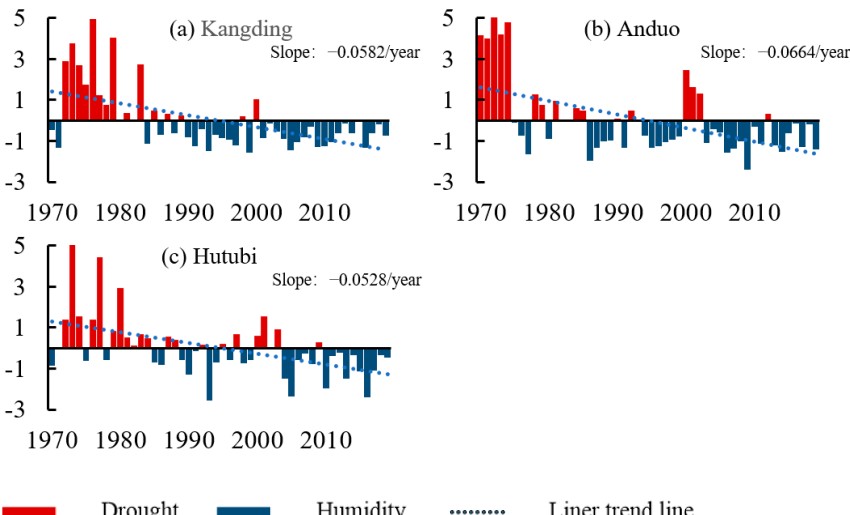

**Figure 4.** Time series of top 5% stations with significant decreases in annual PDI in (**a**) region I, (**b**) region II, and (**c**) region III.

Annual PDI of 703 (85%, 703 out of 830) stations showed statistically significant correlations with annual $SAI_T$ (see Figure 5a). Only five of them showed negative correlations, all of which were in region I. The rest of them showed positive correlations, region I, II, and III had 318 (83%, 318 out of 383), 222 (85%, 222 out of 261), and 158 (85%, 158 out of 186) stations, respectively (see Figure 5a). Slightly dissimilarly, annual PDI showed statistically significant negative correlations with annual $SAI_P$ at all stations (see Figure 5b). There were 651 (78%, 651 out of 830) stations that showed positive relationships between annual $SAI_T$ and $SAI_P$ throughout China, but only at about 29% (188 out of 651) of them were significant ($p < 0.05$) (see Figure 5c). There are 179 (22%, 179 out of 830) stations that showed negative relationships between annual $SAI_T$ and $SAI_P$, but only at about 4% (8 out of 179) of them were significant ($p < 0.05$) (see Figure 5c).

About 71% (592 out of 830) of stations studied in China experienced more drought events (PDI > 1) than humidity events (PDI < −1) during 1970–2019, 14% (117 out of 830) of stations experienced more humidity events than drought events, and 15% (121 out of 830) of stations experienced equal events of drought and humidity (see Figure 6). Meanwhile, 70% (269 out of 383) of stations in region I experienced more drought events than humidity events, 14% (53 out of 383) of stations experienced more humidity events than drought events, 16% (61 out of 383) of stations experienced equal events of drought and humidity. The corresponding data for region II and region III are 70% (184 out of 261), 16% (41 out of 261), and 14% (36 out of 261) and 75% (61 out of 186), 12% (23 out of 186), and 13% (24 out of 186), respectively (see Figure 6).

The drought events (PDI > 1) were experienced at all stations studied, with a frequency range from 6% to 32%. The highest number was seen at the Nanao station in region I, Southeast China (see Figure 7a). In addition, all stations also experienced humidity events with a frequency range from 8% to 36%. The highest number was seen at two stations, the Anduo station in region II, West China, and the Luan station in region I, South China (see Figure 7b). On the other hand, most of the stations experienced extreme drought events (PDI > 3), with a range from 2% to 10%, except at all 43 stations with no historical extreme drought experience. The highest number was seen at the Anduo station in region II, West China (see Figure 7c). However, fewer stations experienced humidity events, only 177 stations showed 2–4% frequency of humidity events (see Figure 7d).

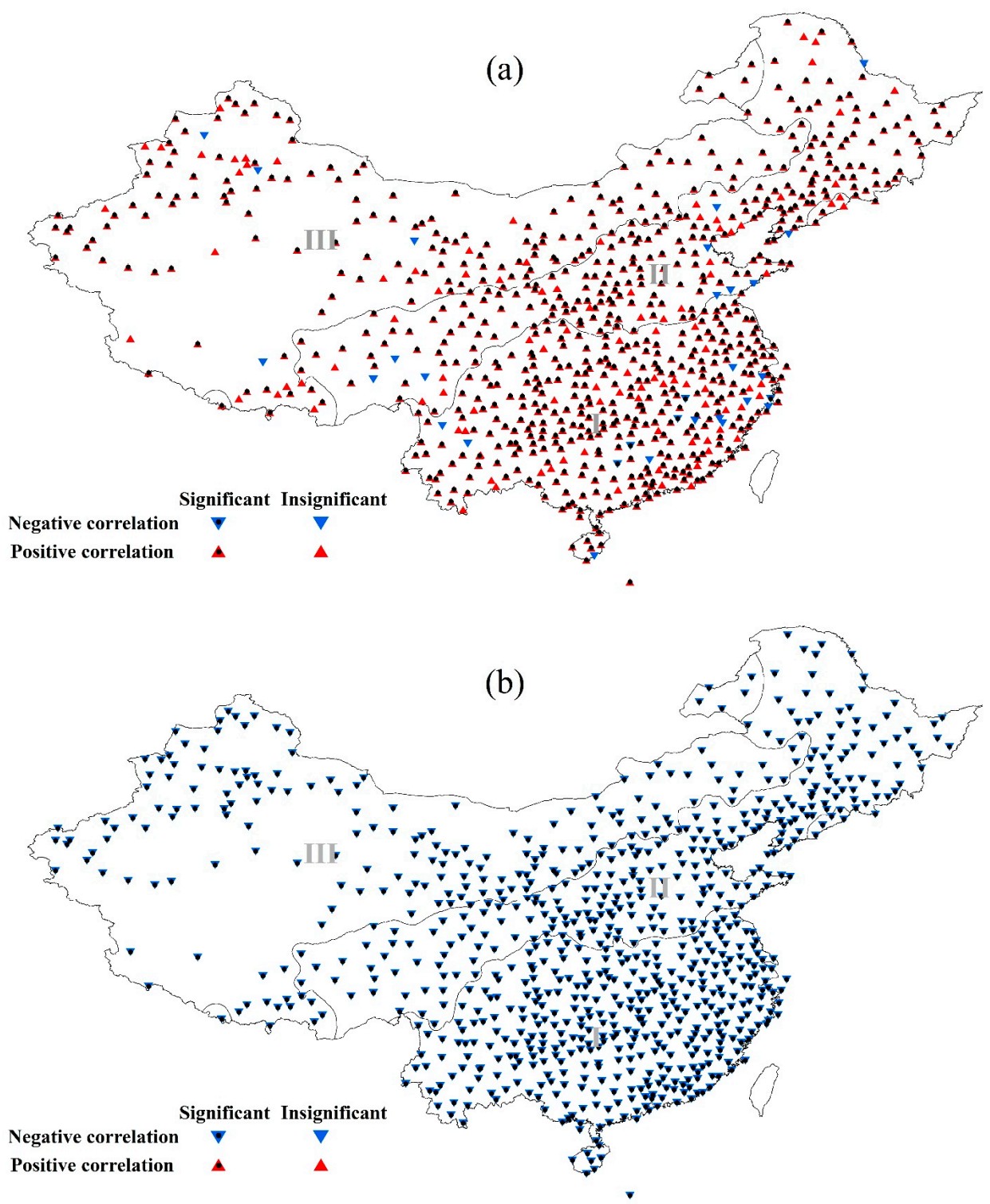

**Figure 5.** *Cont.*

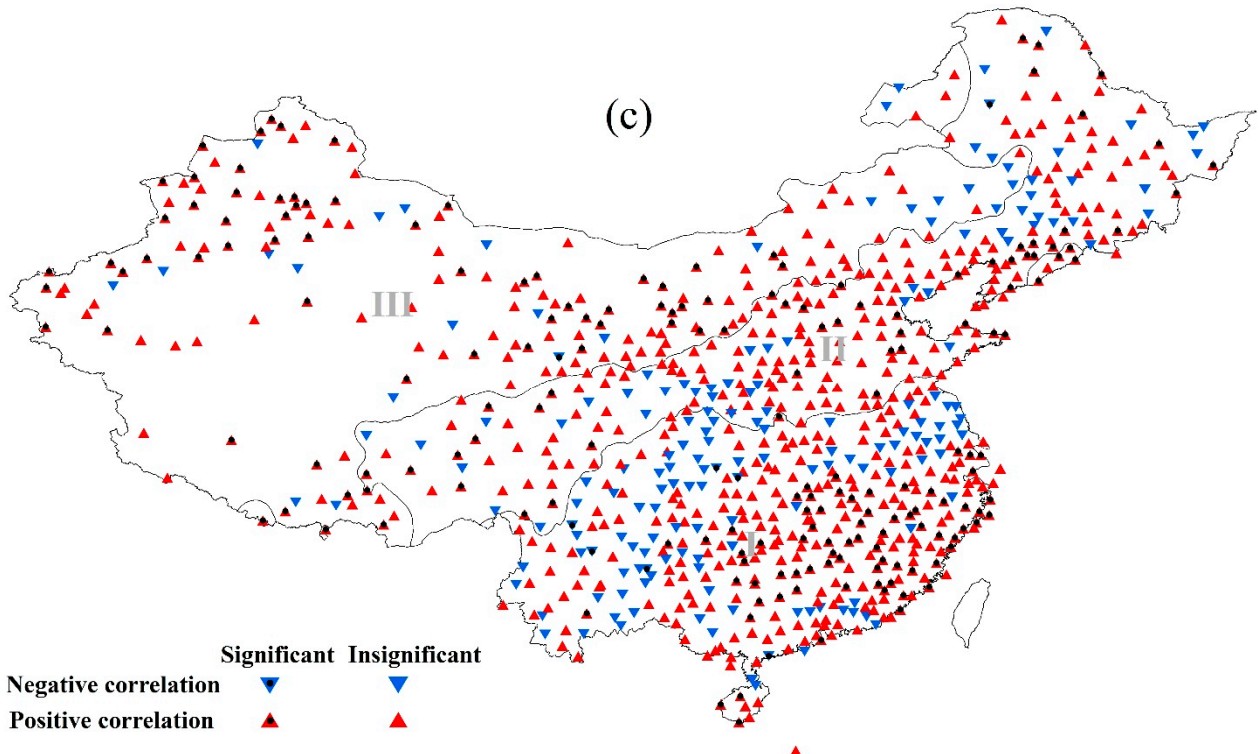

**Figure 5.** Spearman's rank correlations of (**a**) PDI vs. SAI_T, (**b**) PDI vs. SAI_P, and (**c**) SAI_T vs. SAI_P at all stations in China selected by the presented research.

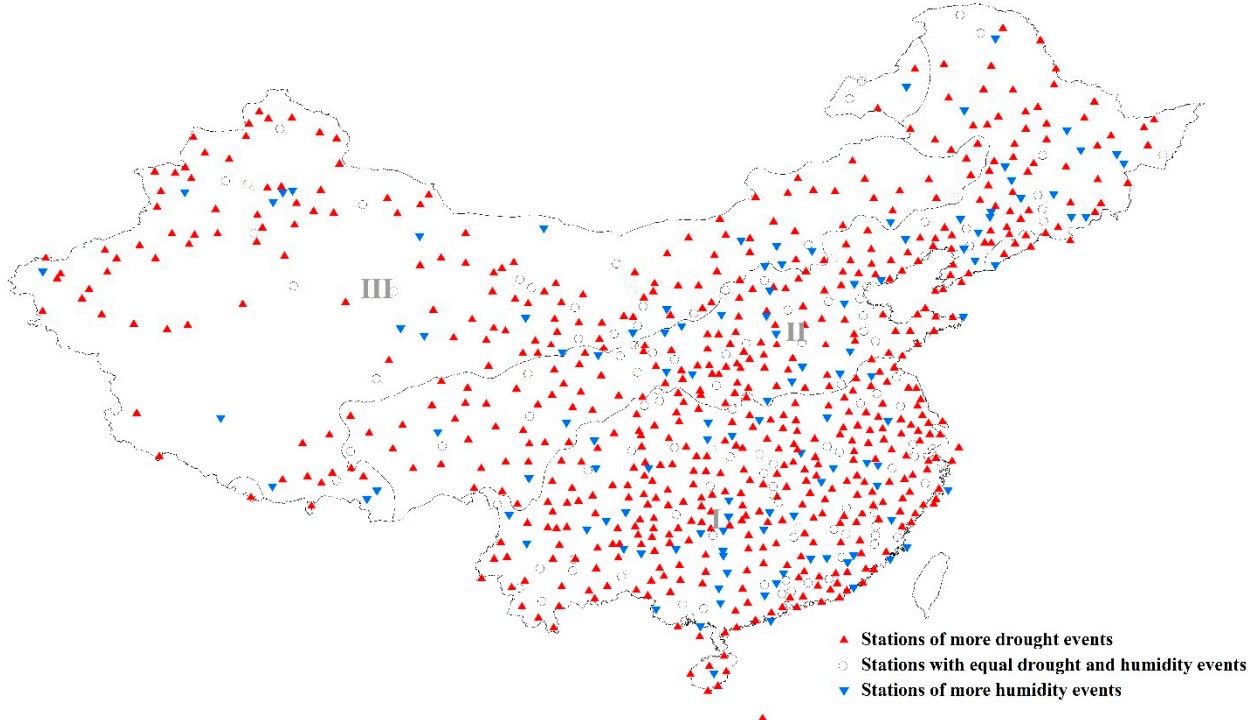

**Figure 6.** Comparison chart of drought and humidity events at all stations in China selected by the presented research during 1970–2019.

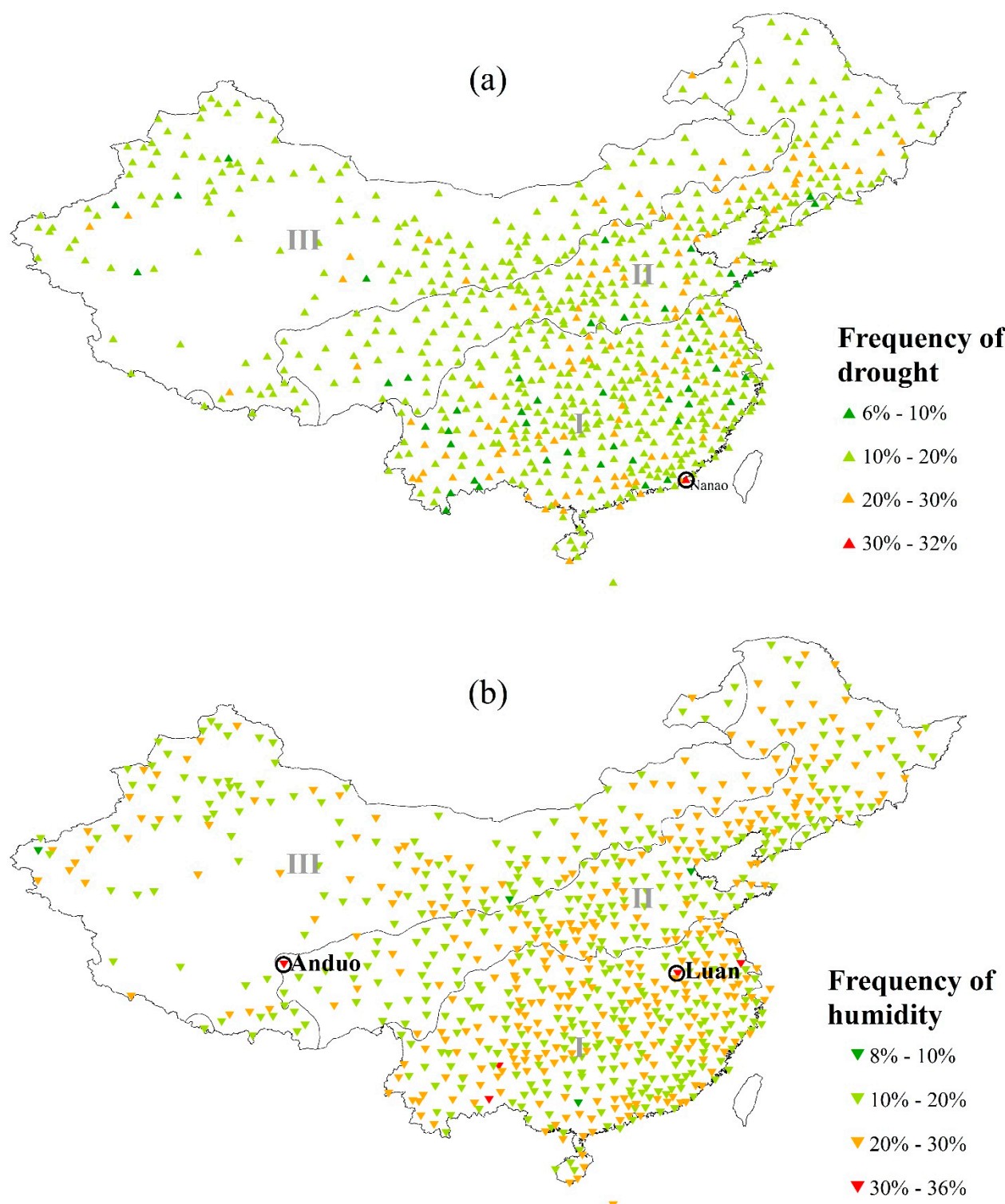

**Figure 7.** *Cont.*

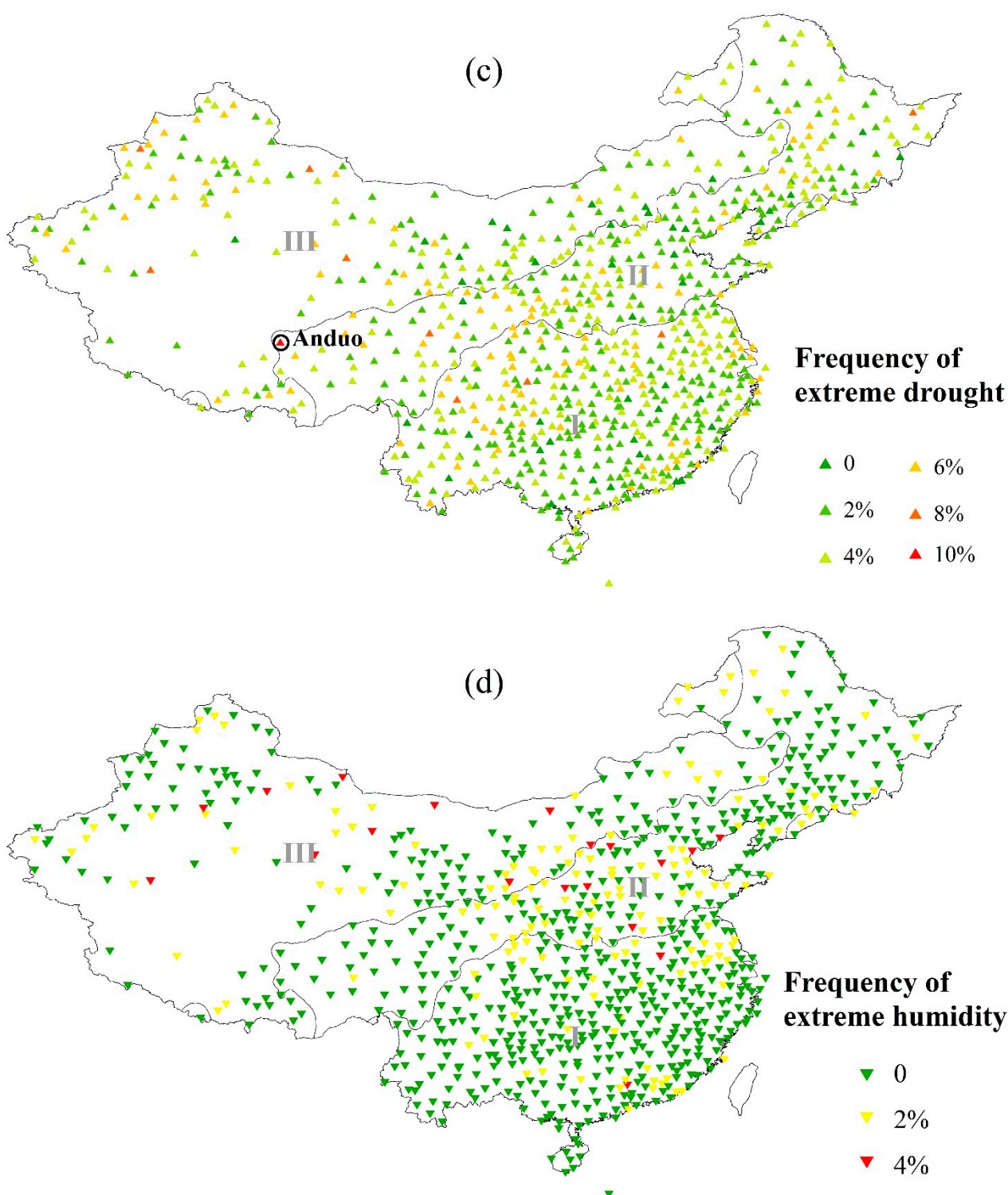

**Figure 7.** Frequency of (**a**) drought (PDI > 1), (**b**) humidity (PDI < –1), (**c**) extreme drought (PDI > 1), and (**d**) extreme humidity (PDI < –3) at all stations in China selected by the presented research during 1970–2019.

The longest drought (humidity) periods were determined based on the consecutive annual PDI values permanently ≥1 (≤−1) at all stations in China during 1970–2019 (see Figure 8). The longest drought duration was more than 5 years, observed at 13 stations (see Figure 9), namely, Wuyiling (1970–1976), Balikun (1970–1975), Wutaishan (2000–2006), Suide (2000–2006), Lishi (2000–2005), Taigu (2000–2007), Jiexiu (2000–2005), Kangding

(1972–1976), Guangyuan (1970–1975), Fuyang (1972–1977), Hangzhou (2009–2014), Nanao (2013–2018), and Dianbai (1972–1977). In addition, the longest humidity duration was more than 5 years observed at 10 stations (see Figure 10), namely, Pulan (1984–1989), Ruoergai (1986–1991), Luolong (1975–1984), Zuogong (1975–1980), Lushi (1986–1991), Qijiang (1977–1982), Xishui (1978–1983), Xianyou (1989–1994), Zhongshan (1976–1982), and Haikou (2014–2019).

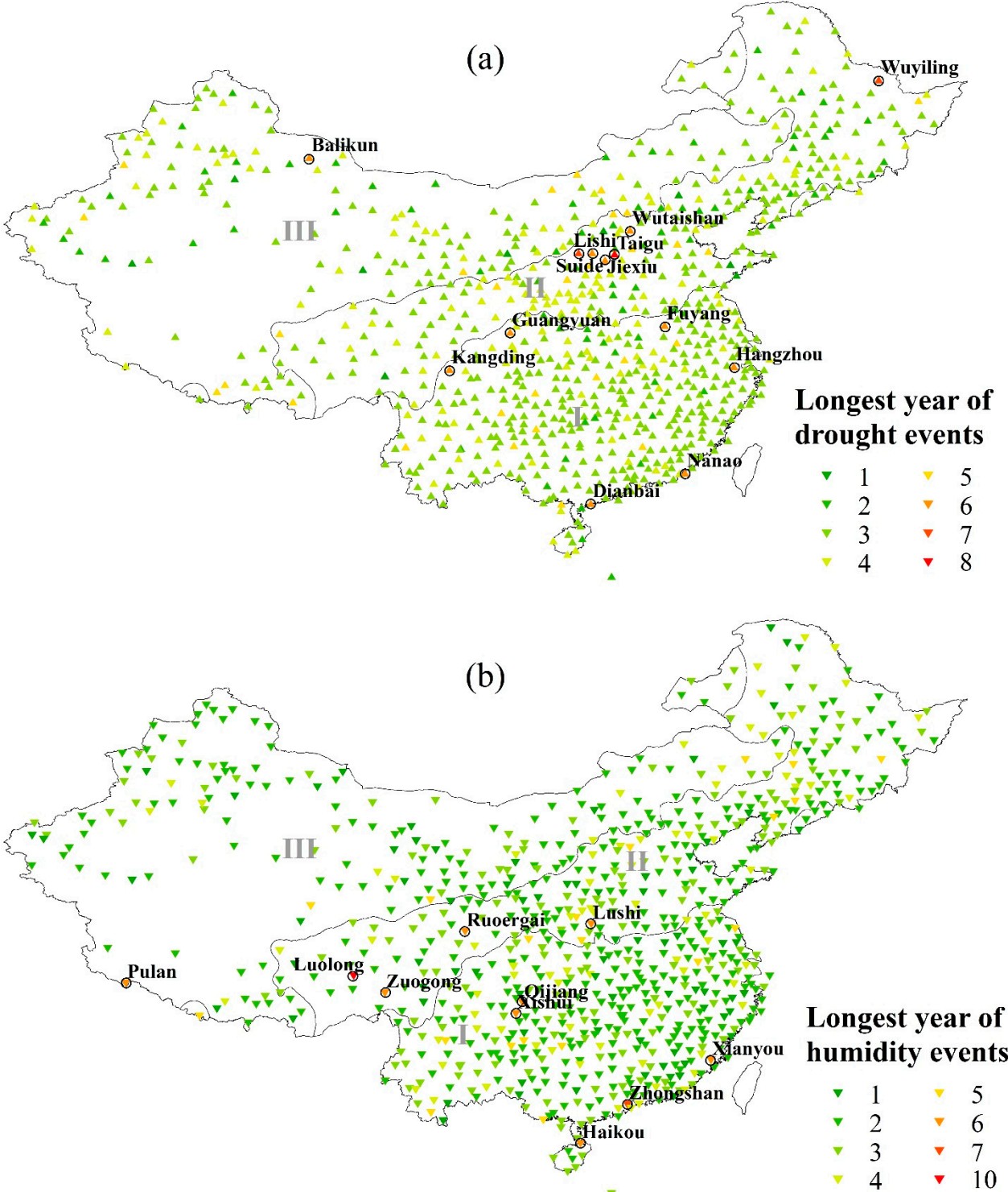

**Figure 8.** Longest year of (**a**) drought (PDI > 1) events and (**b**) humidity (PDI < −1) events at all stations in China selected by the presented research during 1970–2019.

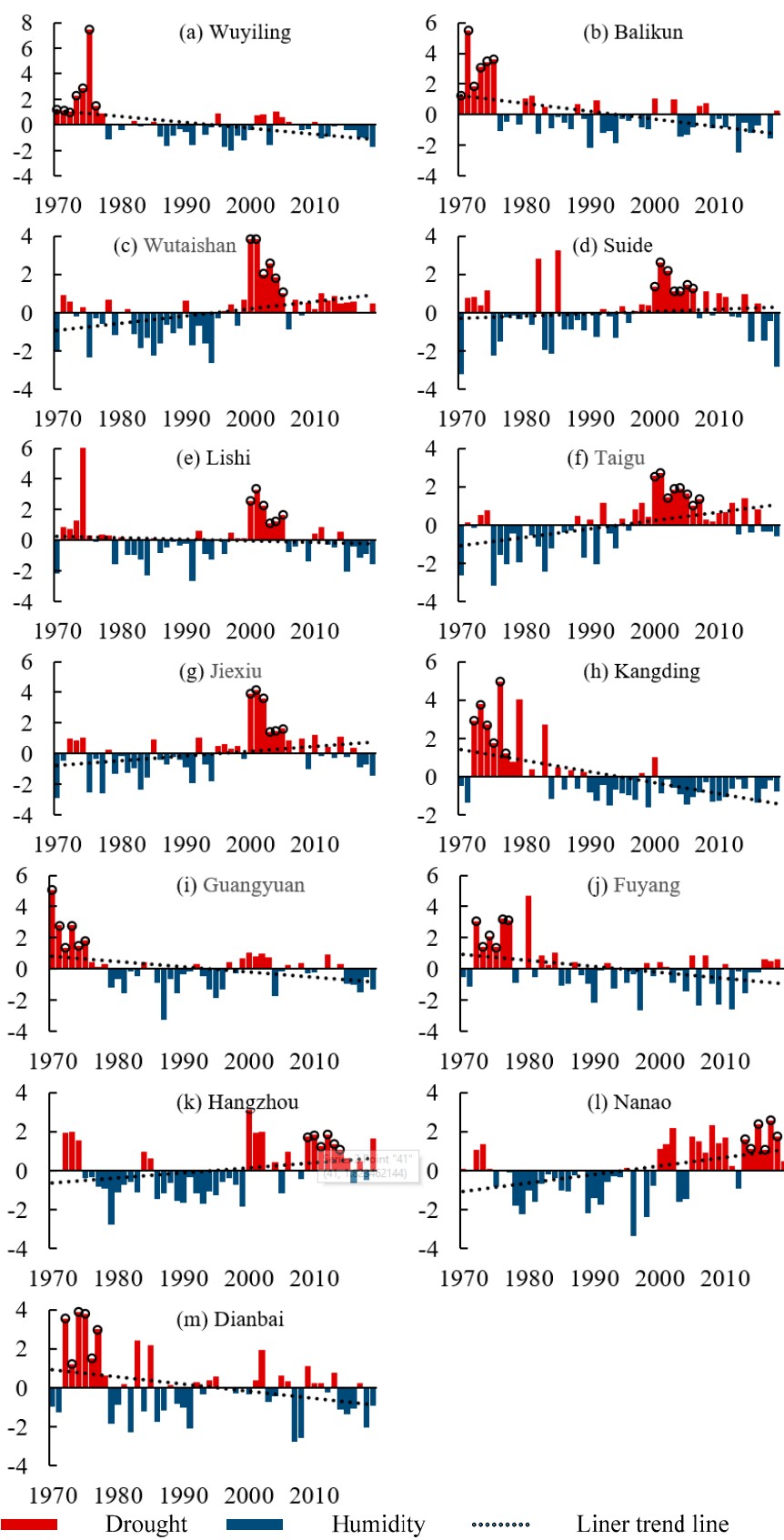

**Figure 9.** Time series of the stations with the longest drought duration lasting more than 5 years.

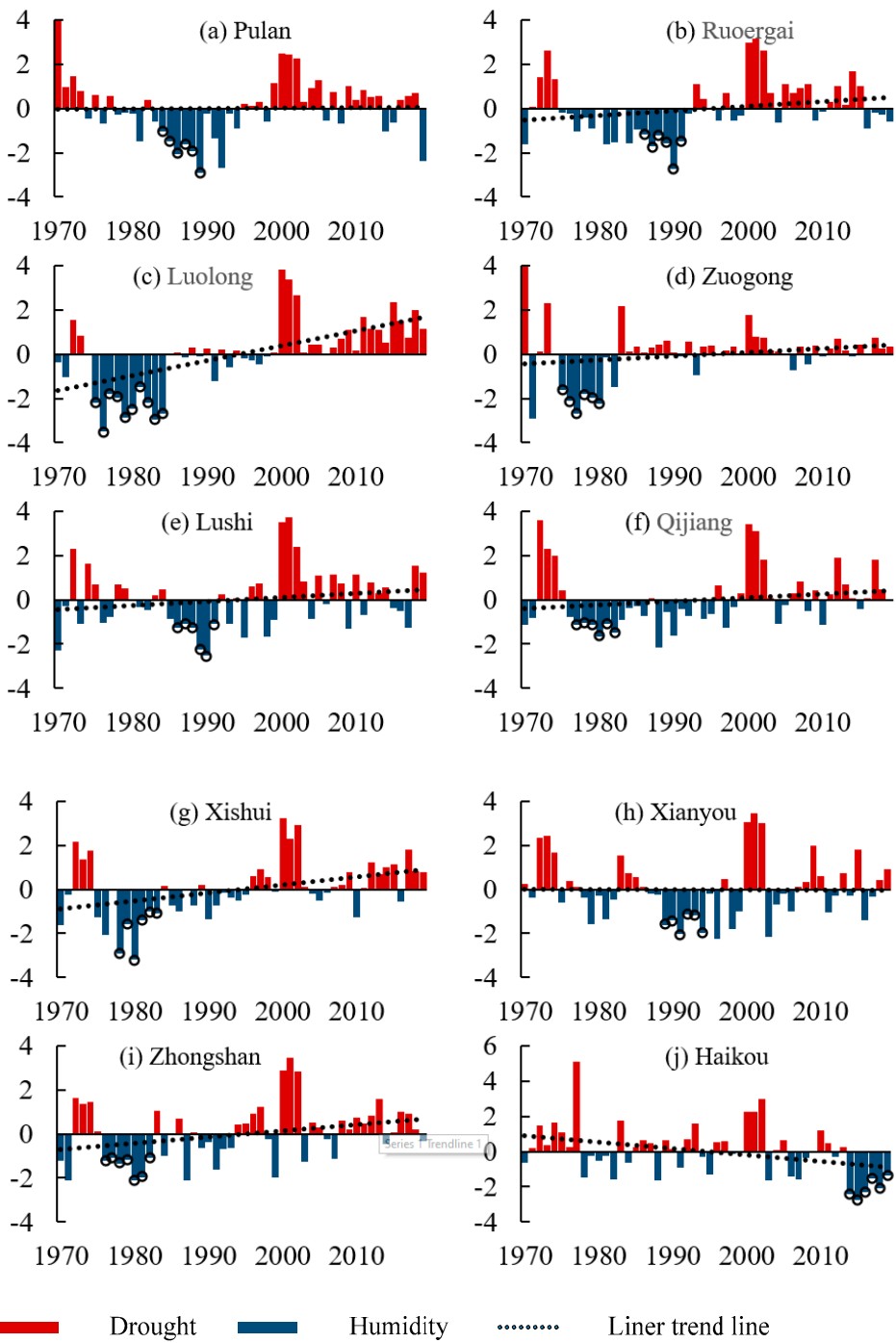

**Figure 10.** Time series of the stations with the longest humidity duration lasting more than 5 years.

The percentage of stations with annual PDI $\geq 1$ and $\leq -1$ represent the spatial distribution of drought and humidity events, respectively, throughout China during 1970–2019 (see Figure 11). More than 67% of stations experienced drought conditions during both periods of 1972–1974 and 2000–2002, even exceeded 80% in the three years 1972, 2000, and 2001 (see Figure 11). None of the stations showed humidity in 1972, while drought events occurred at very few stations during the period of 1989–1991 (see Figure 11). Moreover, both periods of 1976–1983 and 1985–1999 can be considered to a humidity period (drought percentage of stations less than 10%) throughout China (see Figure 11).

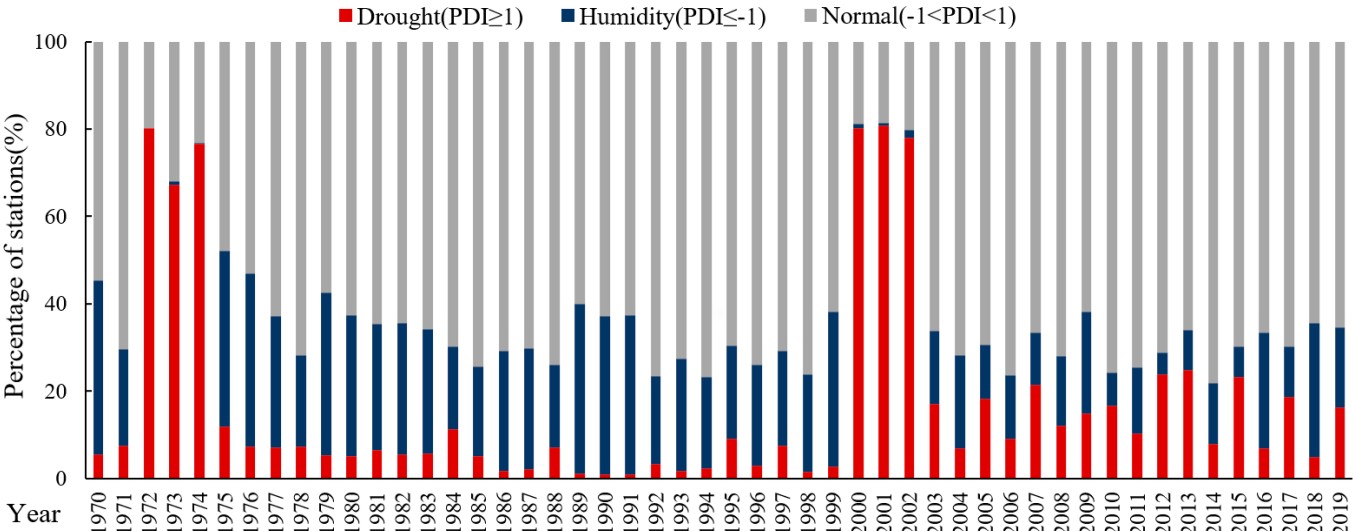

**Figure 11.** Percentage of drought events and humidity events at all stations selected by the presented research during 1970–2019.

## 4. Discussion

The annual PDI time series trend shows that a significant portion of the region has become drier over the past 50 years, with a few areas getting wetter. Meanwhile, the wetter region (region I) had a higher proportion of stations with significant increases in annual PDI, and the drier region (region III) had a higher proportion of stations with the significant decreases in annual PDI. This reveals that there is a tendency toward greater wetness in arid areas, while the characteristics of climate change in humid areas were the opposite throughout China. This is consistent with the drought analysis using SPEI [3]. It is generally believed that the increase in drought trend (annual PDI trend increases) is associated with a significant increase in $SAI_T$ [25], which is consistent with the conclusion of this paper. However, there is no significant correlation between this drought trend (annual PDI trend increases) and $SAI_P$ throughout China. An important reason is that the change of air temperature changes the regional water vapor flux [37,38], which partly suggests that air temperature played a more remarkable role than precipitation in assessing the drying trend. This is consistent with the global analysis of drought with SPEI [39]. Hence, in the context of global warming, air temperature should be given priority in order to assess the drying trend more objectively.

Severity, frequency, duration, and spatial distribution are the four most distinctive features of drought [25,40]. Most regions experienced more drought events than humidity events throughout China during 1970–2019, and the stations with high frequency of drought events were mostly concentrated in the central and eastern regions, while the stations with high frequency of humidity events were scattered. This confirms the results of previous studies by some scholars [3,41–43]. From the results of the statistical analysis, the long-duration drought of the stations was mainly concentrated in two periods, the 1970s and after 2000, this is consistent with the conclusions in a recent report [44,45]. However, the long-duration humidity of stations mostly concentrated in the 1980s and 1990s, this is consistent with the global analysis of drought using SPEI [39]. Additionally, China experienced extensive droughts during two periods, i.e., 1972–1974 and 2000–2002, with a relatively wet period between them, which confirms the conclusions of some research [46–48]. For the period after 2000, the present study indicates that a large part of China has experienced drought that has lasted to this day, which has been shown in recent research [24,39,49].

In present research, PDI was adopted to study the variation characteristics of drought and humidity in China during 1970–2019, and was in good agreement with the results of other drought indices [3,39,41,50]. Therefore, the present research shows that the PDI, like

other drought indices, successfully expresses the interannual variation characteristics of drought and humidity events throughout China and provides certain reference significance for the research and early warning of drought and flood disasters in China.

## 5. Conclusions

The present article employs the annual PDI with the precipitation and temperature data from 830 meteorological stations to systematically study the interannual variation characteristics of drought and humidity in China during 1970–2019. The following major conclusions were drawn:

(1) There were 26.6% of stations that showed significant increases in annual PDI values throughout China during 1970–2019, the proportion of stations with significant increases in region I (31.6%), II (24.1%), and III (19.9%) gradually decreased. Only 6.9% of stations showed significant decreases throughout China, the proportion of stations in region I (3.4%), II (6.5%), and III (14.5%) gradually increased.

(2) Air temperature played a more remarkable role than precipitation in assessing the drying trend with PDI. Both precipitation and air temperature were important in assessing the wetting trend, but the former occupied a more prominent position.

(3) Most stations (85%) showed significant positive correlations between annual PDI and $SAI_T$. PDI and $SAI_P$ were significantly negatively correlated at all stations. $SAI_T$ and $SAI_P$ were positively correlated in some stations (23%) and negatively correlated in a few stations (1%).

(4) Most stations (71%) experienced more drought events, and a few (14%) experienced more wet events or humidity events over the past 50 years. The frequency of drought events ranged from 6% to 32% and humidity events ranged from 8% to 36%. Most stations (95%) experienced extreme drought events, with a frequency range from 2% to 10%. A few (21%) experienced extreme humidity events, with a frequency range from 2% to 4%.

(5) Most stations experienced drought conditions during both 1972–1974 and 2000–2002. Drought events occurred at very few stations during the period of 1989–1991. Both periods of 1976–1983 and 1985–1999 can be considered as humidity periods throughout China.

**Author Contributions:** Data curation, X.W.; formal analysis, X.W.; funding acquisition, X.W. and X.S.; investigation, X.S.; methodology, X.S.; project administration, J.L.; resources, J.L.; supervision, J.L. All authors have read and agreed to the published version of the manuscript.

**Funding:** The research is funded by the Ningxia Key Research and Development Program (Special Talents) (grant no. 2019BEB04029), the Natural Science Foundation of Ningxia (grant no. 2021AAC03043), the First-class Discipline Construction Project of Ningxia University (grant no. NXYLXK2021A03), the Training Project for the Top Young Talents in Ningxia (grant no. 030103030008).

**Institutional Review Board Statement:** Not applicable.

**Informed Consent Statement:** Not applicable.

**Data Availability Statement:** The data presented in this study are available on request from the corresponding author.

**Acknowledgments:** The writers would like to appreciate the editors and the anonymous reviewers for their insightful suggestions to improve the quality of this paper. The writers also acknowledge the assistance of anonymous reviewers.

**Conflicts of Interest:** The authors declare that they have no known competing financial interests or personal relationships that could have appeared to influence the work reported in this paper.

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
