# Peer review of "Spatiotemporal Distribution of Drought and Humidity in China Based on the Pedj Drought Index (PDI)"

_sustainability, doi:10.3390/su14084546_

Round 1

Reviewer 1 Report

General comments

This study presents a very interesting topic based on spatiotemporal drought variability research throughout China using PDI (Pedj drought index). The study uses data from 830 weather stations, which although limited in number compared to the size of the investigated surface, can provide a good indication on the trend of the precipitation and the temperature over the time. 

The study reports a detailed description about the results, but probably a more detailed description about the climatology of the different areas should be reported in a specific section, to give more information about the effects of climate change.

Specific comments

Section 2.2. Data - The authors report: "The missing values in each station were filled with the mean values of the 5 closest stations calculated by the Tyson Polygon method."

Question: How many missing values are there?

Table 1

Comment: The table is placed on two different pages; it needs to be reformatted.

Figure 2

Suggestion: it is necessary to increase the resolution of the figure, it is not easy to consult it.

Figure 5

Suggestion: it is necessary to increase the resolution of the figure, it is not easy to consult it. The caption is on a different page.

Figures 6/7/8

Suggestion: it is necessary to increase the resolution of the figures, it is not easy to consult them.

Reviewer 2 Report

Wu et al. systematically investigated the interannual variation of drought and humidity in China during 1970-2019. The study has done a great deal of work and obtained relatively interesting results that can inform the national-scale water resources management in China. The methodology and results are well written. However, the framework of the manuscript still leaves much room for improvement.

Major comments:

  1. In the abstract, the authors conclude that PDI successfully captures the interannual variability of drought and wetting events in China. However, the authors' full text actually analyses the interannual variability of drought and humidity without addressing whether and to what extent the PDI index more closely captures the actual draught-humidity condition in China. That is, what is the superiority of the PDI index in characterizing drought and humidity compared to other drought indices? I did not find this in the manuscript. Therefore, it can be argued that the conclusions and the study are biased.
  2. Like the previous point, the PDI is a drought index only calculated based on temperature and precipitation. And the evolution of meteorological drought is closely related to evapotranspiration. Therefore, compared with PSDI, SPEI, etc., the PDI index does not have an advantage. Therefore, I suggest that authors should organize their manuscripts mainly in terms of interannual variability of drought and humidity. One point that should be noted is that the last sentence of the penultimate paragraph in the Introduction talks about the gap. However, the purpose of the study in the last paragraph does not correspond to the gap at all. I would suggest reorganizing the two sections so that they correspond before and after.

Minor comments:

Fig. 1. The title is too brief and it is recommended to add a description of the geographical location of the study area, sub-regions, weather stations, DEM, etc.

Figures 2, 5, 6, and 8. Figures are poorly readable. The legend font is too small and affects the judgment of the results.

Figure 3. Units for Slope are missing. Is it years or 10 years?

Conclusion. Too long. Results and conclusions are not the same. Suggest changing the title to ‘summary and conclusion.’ Or the content can be condensed to make the conclusion what it should be.

At the end of the manuscript, the Author Contributions, Funding, Institutional Review Board Statement, Informed Consent Statement., etc., are incomplete.

Round 2

Reviewer 1 Report

The authors have made improvements to the paper.

Figures 2,5,7,8 still need to be improved: they are not easy to interpret because they are too small and have a low resolution.

Author Response

Figures 2,5,7,8 still need to be improved: they are not easy to interpret because they are too small and have a low resolution.

Response:

The authors do appreciate this evaluation and thank for your suggestions. We enlarged the size of figures 2, 5, 7, and 8 and increased the resolution of them in the revision manuscript. Please see attached PDF version.

Reviewer 2 Report

The author has responded well. I think the current manuscript meets the journal's requirements.
One small suggestion.
i) It is not necessary for the manuscript to have all the figures in the text and figures be accurate to the fourth decimal place.
ii) When converting WORD to PDF, choose a high-quality print mode or manually setting a higher resolution can help to improve the readability of the images.

Author Response

i) It is not necessary for the manuscript to have all the figures in the text and figures be accurate to the fourth decimal place.

Response: The authors do appreciate this evaluation and thank for your suggestions. We eliminated the figures in the text after the fourth decimal place, and all the figures are accurate to the fourth decimal place.

ii) When converting WORD to PDF, choose a high-quality print mode or manually setting a higher resolution can help to improve the readability of the images.

Response: The authors do appreciate this evaluation and thank for your suggestions. We increased the resolution of the images in the revision manuscript and generated a high-quality version for your review (please see attached PDF).
